

# Discovery and validation of molecular patterns and immune characteristics in the peripheral blood of ischemic stroke patients

Lin Cong, Yijie He, Yun Wu, Ze Li, Siwen Ding, Weiwei Liang, Xingjun Xiao, Huixue Zhang and Lihua Wang

Department of Neurology, The Second Affiliated Hospital of Harbin Medical University, Harbin Medical University, Harbin, China

## ABSTRACT

**Background**. Stroke is a disease with high morbidity, disability, and mortality. Immune factors play a crucial role in the occurrence of ischemic stroke (IS), but their exact mechanism is not clear. This study aims to identify possible immunological mechanisms by recognizing immune-related biomarkers and evaluating the infiltration pattern of immune cells.

**Methods**. We downloaded datasets of IS patients from GEO, applied R language to discover differentially expressed genes, and elucidated their biological functions using GO, KEGG analysis, and GSEA analysis. The hub genes were then obtained using two machine learning algorithms (least absolute shrinkage and selection operator (LASSO) and support vector machine-recursive feature elimination (SVM-RFE)) and the immune cell infiltration pattern was revealed by CIBERSORT. Gene-drug target networks and mRNA-miRNA-lncRNA regulatory networks were constructed using Cytoscape. Finally, we used RT-qPCR to validate the hub genes and applied logistic regression methods to build diagnostic models validated with ROC curves.

**Results**. We screened 188 differentially expressed genes whose functional analysis was enriched to multiple immune-related pathways. Six hub genes (ANTXR2, BAZ2B, C5AR1, PDK4, PPIH, and STK3) were identified using LASSO and SVM-RFE. ANTXR2, BAZ2B, C5AR1, PDK4, and STK3 were positively correlated with neutrophils and gamma delta T cells, and negatively correlated with T follicular helper cells and CD8, while PPIH showed the exact opposite trend. Immune infiltration indicated increased activity of monocytes, macrophages M0, neutrophils, and mast cells, and decreased infiltration of T follicular helper cells and CD8 in the IS group. The ceRNA network consisted of 306 miRNA-mRNA interacting pairs and 285 miRNA-lncRNA interacting pairs. RT-qPCR results indicated that the expression levels of BAZ2B, C5AR1, PDK4, and STK3 were significantly increased in patients with IS. Finally, we developed a diagnostic model based on these four genes. The AUC value of the model was verified to be 0.999 in the training set and 0.940 in the validation set.

**Conclusion**. Our research explored the immune-related gene expression modules and provided a specific basis for further study of immunomodulatory therapy of IS.

Corresponding author
Lihua Wang,
wang_lihua0910@163.com

# INTRODUCTION

Stroke is the second leading cause of death and the third leading cause of disability in adults (*Campbell & Khatri, 2020*), and more than 80% of strokes are ischemic (*Hasan et al., 2018*). Ischemic stroke (IS) seriously affects patients' quality of life and causes a heavy burden to families and society. Therefore, our study focuses on discovering novel targets for diagnosing and treating IS.

Currently, IS diagnosis mainly depends on clinical manifestations and neuroimaging procedures, such as computed tomography (CT) (*Harston et al., 2015*) and magnetic resonance imaging (MRI) (*Zameer, Siddiqui & Riaz, 2021*). The most effective early treatment methods are thrombolysis and intravascular therapy (*Cassella & Jagoda, 2017*; *Hasan et al., 2018*), both of which have time limitations when administering. The clinical manifestations of the disease are variable, and the cost of administering CT and MRI scans is expensive and time-consuming. If the symptoms are not typical or the above examinations cannot be performed promptly, diagnosis and treatment may be delayed. Additionally, the above treatment methods carry the risk of bleeding. This has led us to search for new diagnostics and treatments.

Genetic factors are related to the occurrence and prognosis of stroke (*Malik et al., 2018*; *Torres-Aguila et al., 2019*). Possible stroke-related gene loci, such as ANK2, CDK6, and KCNK3, were found using whole genome sequencing (*Malik et al., 2018*). However, the specific mechanism of action was not clear. CYP450, COX-2, PTGIS, TBXAS1, P2RY1, TGB3, and GPIIa are candidate genes that may be associated with stroke prognosis (*Torres-Aguila et al., 2019*). Presently, bioinformatics technology has been widely applied to various diseases, so we used bioinformatics technology to screen and identify key genes associated with IS and build a classification diagnosis model based on these genes, in order to provide certain auxiliary methods for the diagnosis of IS and potential therapeutic targets for the management of diseases.

Immune factors are also closely associated with IS (*Iadecola, Buckwalter & Anrather, 2020*; *Levard et al., 2021*). In the acute phase, the immune response is activated immediately when the brain is in ischemia. Damaged or dead brain cells release damage-associated molecular patterns (DAMPS) (*Wu et al., 2022*). These substances quickly activate intracranial immune cells (such as microglia), release chemokines and cytokines, and trigger a series of inflammatory cascades (*Anrather & Iadecola, 2016*; *Levard et al., 2021*). More importantly, immune cells in the peripheral blood can penetrate the central nervous system (CNS) through the damaged blood–brain barrier (BBB) (*Jayaraj et al., 2019*), further exacerbating damage. At the same time, these DAMPS and cytokines can enter the bloodstream and activate the systemic immune system and inflammatory response, which can lead to severe immunosuppression and even fatal infections (*Iadecola, Buckwalter & Anrather, 2020*). In the chronic phase, adaptive immune responses to the brain are

mobilized due to antigen presentation, which may be the basis of neuropsychiatric sequelae, poststroke dementia/fatigue/depression, and the root cause of post-stroke morbidity (*Iadecola, Buckwalter & Anrather, 2020*). Epidemic-modulated therapy may be a good alternative to IS therapy. Several studies have shown that immune regulation can delay disease progression and improve neurological function and prognosis. These changes highlight the need to maintain a stable immune microenvironment in the CNS (*Javidi & Magnus, 2019*; *Jayaraj et al., 2019*). TGF-$\beta$1 has a significant neuroprotective effect (*Taylor et al., 2017*) and can reduce post-stroke infection (*Cekanaviciute et al., 2014*; *Doll, Barr & Simpkins, 2014*). Relevant targeted drugs are now being developed. Specific inhibitors of IL-1$\beta$ can delay atherosclerosis by interfering with immune pathways associated with atherosclerotic plaque formation (*Khambhati et al., 2018*). Meanwhile, IL-1 receptor antagonists have been found to effectively reduce peripheral inflammation in patients with acute IS and improve clinical outcomes in these patients (*Smith et al., 2018*). In conclusion, the analysis of immune infiltration analysis and study of the relationship between target genes and immune cell distribution may help to elucidate the immune-related molecular mechanism of IS and provide the specific basis for immunoregulatory therapy. At the same time, the prediction of drug target gene action network may provide a new target for the treatment of IS.

Competing endogenous RNA (ceRNA) does not represent a specific type of RNA, but a regulatory mechanism. By combining with microRNA response elements (MREs) existing on mRNA, miRNAs inhibit mRNA translation or lead to its degradation, thus achieving the function of regulating gene expression after transcription (*Salmena et al., 2011*). Different mRNAs can compete to bind to the same miRNA and thus participate in the regulation of gene expression. Transcriptome studies have found that MREs exist not only on mRNA, but also on lncRNA, circRNA, and other types of RNA, which means that the same miRNA can bind to several and multiple types of RNA. A competitive relationship is formed between RNA molecules that bind to the same miRNA, which forms the ceRNA network. If the regulation of ceRNA is abnormal, it may cause the occurrence of diseases, as shown in the research of tumors (*Karreth et al., 2015*), stroke (*Li, Duan & Fu, 2020*), and other complex diseases. It has been reported that miRNAs intervene in the occurrence and prognosis of stroke by modulating immune and inflammatory factors. *Huang et al. (2018)* found that the inhibition of microRNA-210 suppresses pro-inflammatory response and reduces acute brain injury of IS in mice. *Boldin et al. (2011)* reported that miR-146a inhibits the production of inflammatory cytokines. In addition, the occurrence, progression, and prognosis of acute IS are also related to abnormal lncRNA expression (*Bai et al., 2014*; *Feng, Guo & Ai, 2019*; *Fu et al., 2021*; *Li, Duan & Fu, 2020*; *Wang et al., 2020*). Therefore, they are potential candidates for stroke diagnosis. Meanwhile accumulating evidence demonstrates that lncRNAs regulate target gene expression by acting as ceRNAs for miRNAs and participate in immunoinflammation in cerebral ischemia-reperfusion injury and various diseases. Knockdown metastasis-associated lung adenocarcinoma transcript 1 (MALAT1), a ceRNA for miR-181c-5p, attenuated inflammatory injury after cerebral ischemia (*Cao et al., 2020*). lncRNA SNHG14 promotes inflammatory response induced by cerebral ischemia /reperfusion injury by regulating miR-136-5p/ROCK1 (*Zhong, Yu & Qin,*

*2019*). The lncRNA small nucleolar RNA host gene 15 (SNHG15) was shown to regulate the expression of programmed death ligand 1 (PD-L1) by inhibiting miR-141, which in turn promoted the resistance of stomach cancer cells to the immune response (*Dang et al., 2020*). The lncRNA HIX003209 functioned as a ceRNA and enhanced inflammation by sponging miR-6089 *via* the Toll-like receptor 4 (TLR4)/nuclear factor (NF)-$\kappa$B pathway in macrophages in rheumatoid arthritis (*Yan et al., 2019*). The construction of a target gene-miRNA-lncRNA-based ceRNA network may provide a molecular basis for understanding stroke and help predict the possible immunological pathways of IS.

In this study, the least absolute shrinkage and selection operator (LASSO) regression and support vector machine-recursive feature elimination (SVM-RFE) algorithm were applied to screen essential genes, and their relationship was subsequently analyzed with immune cells. The combined use of two algorithms made the process more rigorous and standard. A ceRNA network of hub genes-miRNA-lncRNA was then established. Finally, a logistic regression diagnostic model was built based on hub genes to provide an auxiliary method for the diagnosis of IS. The technical route is shown in Fig. 1.

## MATERIALS & METHODS

### Obtaining and processing profile datasets

We obtained and downloaded the IS gene expression profile from the Gene Expression Omnibus (GEO, http://www.ncbi.nlm.nih.gov/geo) (Table 1). The criteria for screening were that the data in the dataset contained both IS and control groups (CTL), and the number of cases in each group was no less than 20. Patients in the IS group were those diagnosed with acute IS. In addition, the data were based on whole blood samples with full gene expression profiles. After screening, GSE16561 (*Barr et al., 2010*) and GSE58294 (*Stamova et al., 2014*) matched the criteria. GSE16561 was used as a training set. GSE58294 was used for the validation of biomarker genes. We applied the Perl language script (https://www.perl.org/) to get the gene name corresponding to each probe from the platform annotation file, and the names were then converted. In this process, we used the average of all probes' expressions if there were multiple probes corresponding to the same gene name.

### Differential expression analysis

We applied R package limma (*Ritchie et al., 2015*) (version 3.6.0) to perform differential expression analysis to identify DEGs between groups in the GSE16561 dataset. FoldChange >1.5 and adjusted *P*-value < 0.05 were taken as significant criteria. DEGs were represented by heatmap, designed by the pheatmap package (*Kolde, 2012*) (version 1.0.12), and volcano maps, created by the ggplot2 package (*Wickham, 2009*) (version 3.3.5).

### Function enrichment analysis

We not only annotated the function of the differentially expressed genes themselves using Gene Ontology (GO) enrichment, but also studied the pathways through Kyoto Encyclopedia of Genes and Genomes (KEGG) analysis. GO Enrichment and KEGG pathways analysis results were obtained using clusterProfiler (*Yu et al., 2012*). GSEA software was used to analyze the potential molecular mechanisms and signaling pathways
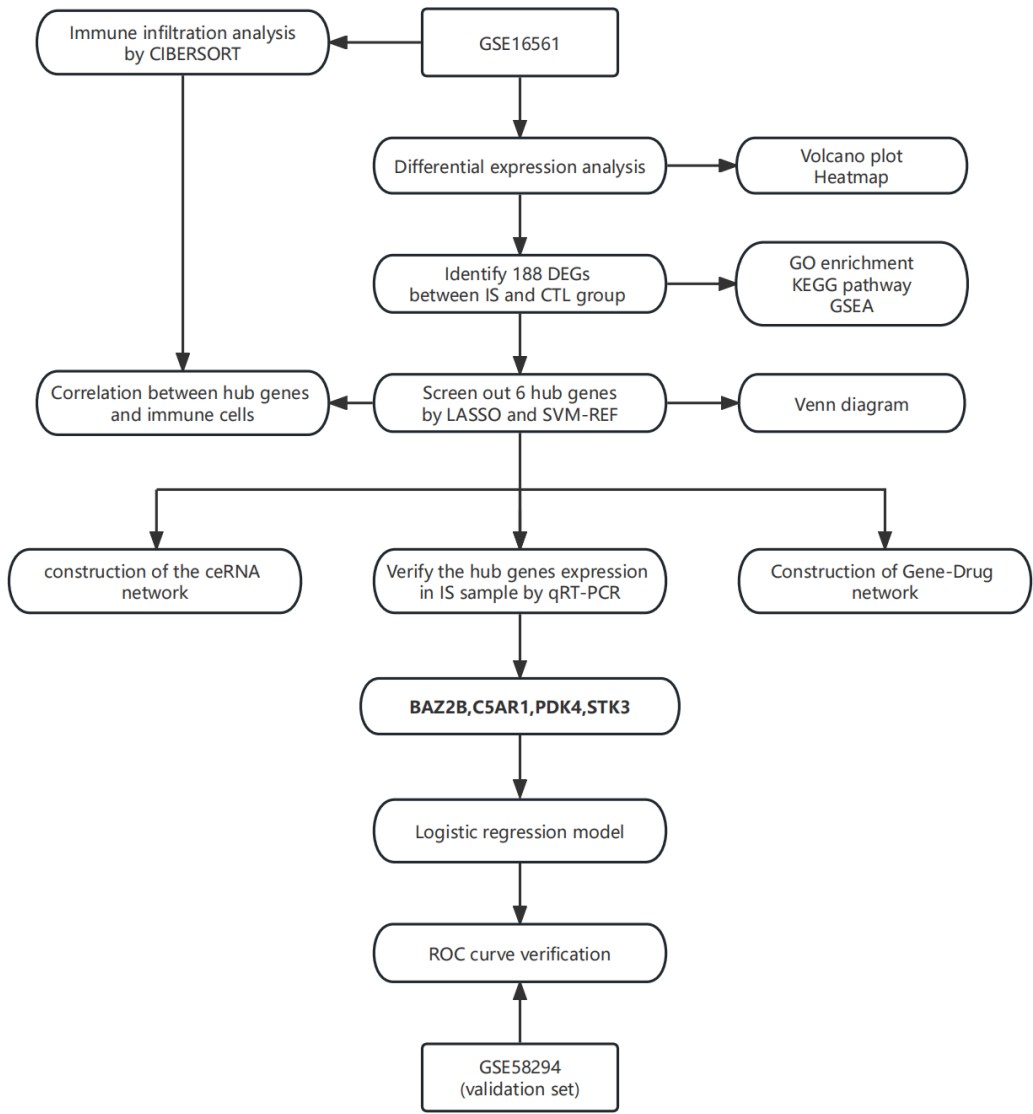

**Figure 1** **The technical route.** Abbreviations: IS, ischemic stroke; CTL, healthy control.

of hub genes in IS patients. C2 (c2.cp. Kegg. v7.4.symbols. gmt) was selected as the reference gene set for analysis. The R package clusterProfiler also conducted the GSEA analysis of the IS and CTL groups, with an adjusted *P*- value < 0.05.

## Screening for hub genes

Two machine learning methods (LASSO and SVM-RFE) were used to identify the hub genes of IS. These two methods were first applied to analyze breast tumors (*Wang & Liu, 2015*), and then also to neurological diseases such as AD (*Liu, Li & Pan, 2021*). LASSO regression is characterized by variable selection and complexity adjustment while fitting a generalized linear model. Thus, the target-dependent/response variable can be modelled and predicted whether it is continuous, binary, or multivariate discrete (*Friedman, Hastie & Tibshirani,*

**Table 1  Basic information on gene expression profiling.**

| Location | Dataset ID | Platform | Type | Number |
|---|---|---|---|---|
| Whole blood samples | GSE16561 | GPL6883 | Microarray | 39IS VS 24CTL |
| Whole blood samples | GSE58294 | GPL570 | Microarray | 69IS VS 23CTL |

*2010*). The LASSO analysis was undertaken using the glmnet package (*Engebretsen & Bohlin, 2019*) with the following parameters: response type set to binomial, alpha set to 1, and 10-fold cross-validation to adjust the optimal value of the parameter λ. SVM has shown many unique advantages in solving small samples and in nonlinear and high-dimensional pattern recognition, and can be extended to other machine learning problems such as function fitting. SVM-RFE, on the other hand, is used as a practical feature selection technique to find the best variables by removing the feature vectors generated by SVM (*Wang & Liu, 2015*). The algorithm was constructed using the e1071 package and halve.above = 100 and $k = 5$ were used as parameter criteria. The hub genes were the intersecting genes obtained by the two machine learning algorithms.

## Evaluation of immune cell infiltration

Analysis of immune cell infiltration relied on CIBERSORT, which is a widespread tool to research the proportion of 22 immune cells in specimens. Furthermore, we analysed the Spearman correlation between immune cells and key genes. The corrplot software package (*Friendly, 2002*) calculated the Pearson correlation coefficient between hub genes and each immune cell.

## Construction of the ceRNA network

First, the miRanda, MicroRNADB, and TargetScan databases were used to simultaneously predict the target miRNAs of hub genes and determine the intersection. We obtained the miRNA-mRNA interaction pair file. Second, the spongeScan database (https://bioinformaticshome.com/tools/rna-seq/descriptions/spongeScan.html#gsc.tab=0) was used to predict the lncRNA based on the resulting file from the previous step. Finally, the ceRNA network was formed, and the feature files were jointly built based on the lncRNA-miRNA-mRNA files. We visualised ceRNA regulatory networks using Cytoscape (*Shannon et al., 2003*).

## Gene-drug target network analysis

The Drug-Gene Interaction Database (DGIDB, http://www.dgidb.org/), an online database that collects data on drug and gene interactions, was the source of material for building the network. The hub gene-drug interaction pair was compared with the DGIDB database to identify immunomodulatory drugs that may be related to IS. The gene-drug target network was constructed based on the above information.

## Sample collection

Our samples were all obtained from patients in the Department of Neurology of the Second Affiliated Hospital of Harbin Medical University, approved by the Medical Ethics

Committee of the Second Affiliated Hospital of Harbin Medical University (NO. KY2022-285), and all enrolled individuals signed an informed consent form. There were five cases in the disease group and five cases in the healthy CTL. The disease group's (IS group) inclusion criteria: patients were suffering from their first acute IS confirmed by MRI, meeting the diagnostic criteria in the International Classification of Diseases (9th Revision). Exclusion criteria: patients with a history of hematologic disorders, type 1 diabetes, autoimmune, thyroid, neoplastic, renal, or liver diseases were excluded. We collected 5 ml fasting venous blood samples across both groups. Peripheral blood for the disease group was required to be collected within 24 h of onset.

### Real-time quantitative PCR (RT-qPCR)

RT-qPCR was used to detect the expression of six hub genes in the peripheral whole blood clinical samples. Total RNA was isolated from each sample using a Trizol reagent according to the manufacturer's instructions. Reverse transcription from RNA to cDNA was performed using HiScript Q RT SuperMix for qPCR (+gDNA wiper) (Vazyme Biotech, Nanjing, China). RT-qPCR was performed on cDNA samples using ChamQ SYBR Color qPCR Master Mix (2X) (Vazyme Biotech, Nanjing, China). Results were analyzed using the 2-$\Delta\Delta$Ct method and expressed as fold changes, with GAPDH selected as the internal reference gene. The PCR primers were designed by Majorbio (Shanghai, China) (Table S1). A $P$-value less than 0.05 was considered statistically significant.

### Statistical analysis

The Shapiro–Wilk test was used to assess whether the continuous data were normally distributed in the CTL and IS groups. Normally distributed data were counted using the independent samples $t$-test. Non-normally distributed data were evaluated using the Wilcoxon-Mann Whitney test. A $P$-value < 0.05 was considered statistically significant.

### Development and validation of a diagnostic model

Boxplots were used to show the expression of hub genes in GSE58294 as a validation dataset. We considered the difference statistically significant at a $P$-value < 0.05. A logistic regression algorithm was used to build a diagnostic model for IS classification based on crucial genes. The Proc R package was applied to generate the ROC curve and calculate the area under the ROC curve (AUC) (*Stamova et al., 2014*) to verify the accuracy of the genes and the model in dataset GSE16561. In addition, we calculated the AUC values of the key genes and logistic regression models in dataset GSE58294 to validate the diagnostic models' classification performance. To avoid the occurrence of overfitting, we chose to reduce the complexity of the model and reduce the parameters to make the model simpler. When training the model, the dataset was partitioned in a 7:3 ratio for training the model and predicting the accuracy of the created model, respectively, to balance the accuracy. Finally, the visual analysis of the above process was completed using ggplot2.
## RESULTS

### Differential expression analysis

The volcano plot (Fig. 2A) showed that by using R package limma, we obtained 188 differential expression genes (Table S2), 85 genes of which were down-regulated in expression and 103 were up-regulated in expression. The top 50 differentially expressed genes are shown in a heatmap (Fig. 2B) created by the pheatmap package.

### Function enrichment analysis

The GO analysis results (Table S3) were presented in a barplot (Fig. 3A), a bubble plot (Fig. 3B), and a circle plot (Fig. 3C). GO-Biological Process (BP) shows that DEGs were primarily involved in immune-related biological processes, including immune response-regulating signaling pathway, positive regulation of cytokine production, and immune response-regulating cell surface receptor signaling pathway; GO molecular function (MF) enrichment results showed DEGs were primarily associated with integrin binding oxidoreductase activity, acting on NAD(P)H quinone or a similar compound as acceptor and immunoglobulin binding; and GO-cellular component (CC) analysis was significantly enriched in platelet alpha granule and secretory granule lumen. KEGG pathways of each module (Fig. 4 and Table S4) were mainly enriched in the MAPK signaling pathway, cell adhesion molecules, complement and coagulation, and the cascades. The GO and KEGG pathway analysis results suggested that the immune system played a crucial role in IS.

We further conducted a GSEA analysis to avoid missing the functions of some insignificant but biologically significant genes, biological characteristics, and regulatory networks. GSEA results (Fig. 5, Table S5) showed that the pathway involved axon guidance, complement and coagulation cascades, Fc gamma R-mediated phagocytosis, focal adhesion, and regulation of actin cytoskeleton in the IS group (Fig. 5A). Complement and coagulation cascades, the identified pathway of IS, were consistent with the result of KEGG, and the other pathways are related to immunity or cellular metabolism. Genes in the treatment group (IS group) were enriched at the top, which means that this gene set was upregulated. Conversely, genes in the CTL group (Fig. 5B) were enriched at the bottom with a down-regulation trend.

### Identification of the hub gene

To explore the biomarkers of IS, we performed feature screening through LASSO regression and the SVM-RFE algorithm. Twelve genes were identified as the most associated with IS by LASSO regression (Fig. 6A). The SVM-RFE algorithm evaluated 34 characteristic genes in IS (Fig. 6B). Both machine algorithms were subjected to 10-fold cross-validation to ensure the accuracy of the results. Six differential expression genes (ANTXR2, BAZ2B, C5AR1, PDK4, PPIH, and STK3) (Fig. 6C) were then identified as the hub genes by these two joint algorithms for subsequent research.

### Results of immune cell infiltration analysis

The immune cell infiltration analysis results (Table S6) were presented by barplot (Fig. 7A) and heatmap (Fig. 7B). Barplot showed a difference in the percentage of immune cells

A

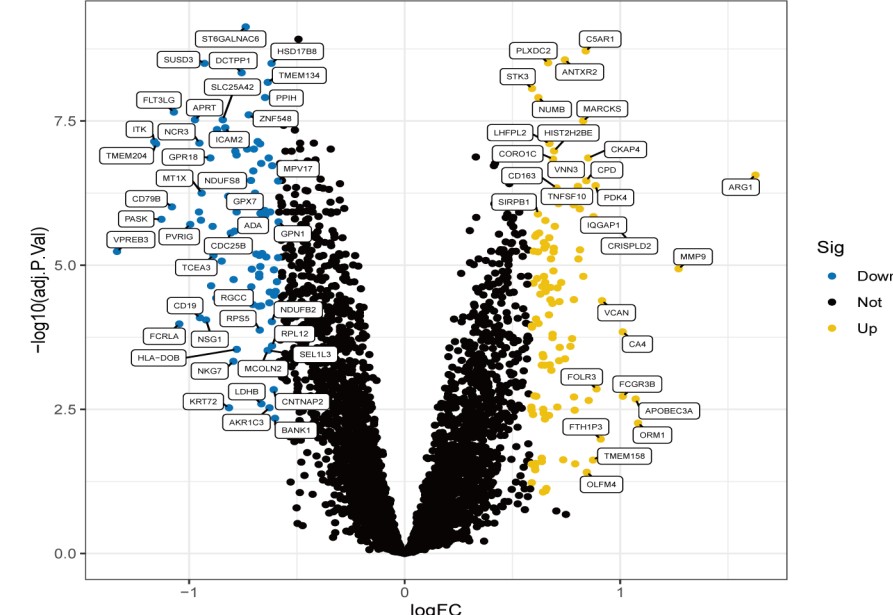

B

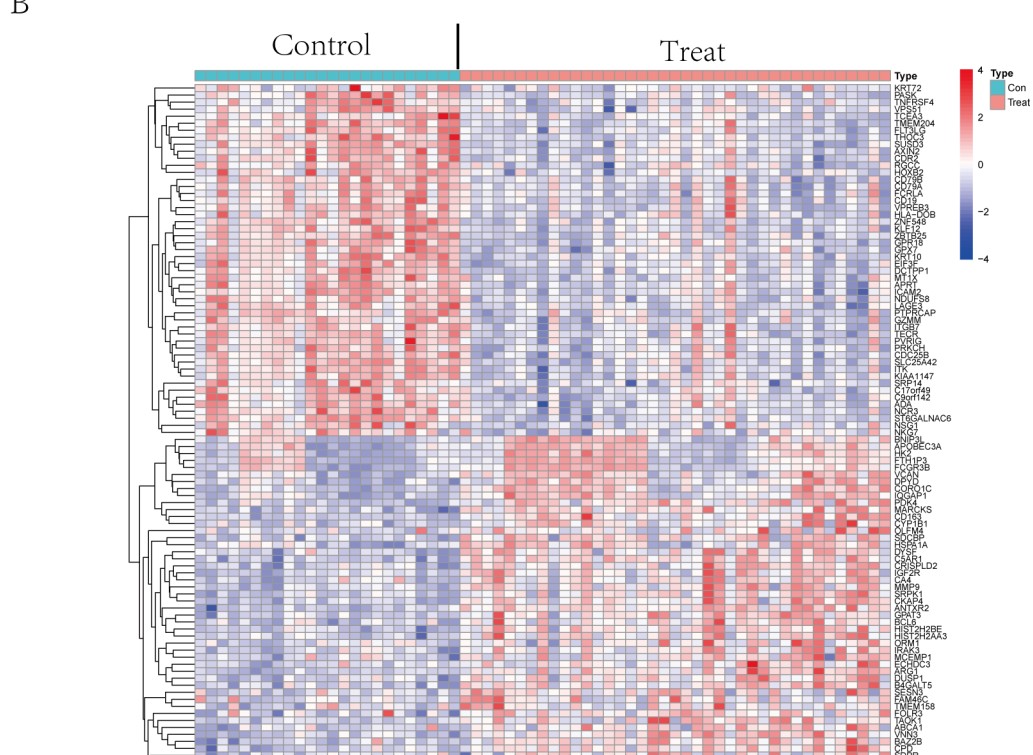

**Figure 2** **Differential expression analysis.** (A) The differential expression analysis results are shown in the volcano plot. The *x*-axis represents log1.5 (fold change), and the *y*-axis represents -log10 (adjust p. value). Blue dots represent downregulated genes, yellow dots represent upregulated genes, and black dots represent genes with no evident differential expression. (continued on next page...)

**Figure 2 (...continued)**
(B) Heatmap of DEGs. Each column in the graph represents a sample, each row represents a gene, and the expression status of the gene is indicated from high to low in orange to blue, respectively, and at the top of the heatmap, red/blue represents the IS group/CTL group, respectively. Abbreviations: ischemic stroke (IS), healthy control (CTL), differentially expressed genes (DEGs).

between the IS group and CTL. Compared with the CTL, monocyte, macrophage M0, neutrophil, and mast cell activated infiltration increased, but T follicular helper cells and CD8 T cell infiltration decreased in the IS group. The relationship between immune cells and key genes is presented in Fig. 8. ANTXR2, BAZ2B, C5AR1, PDK4, and STK3 were positively correlated with neutrophils and negatively correlated with T follicular helper and CD8 T cells, but the PPIH gene was positively associated with T follicular helper and CD8 T cells and negatively correlated with mast cell activation and neutrophils. Only PPIH was a down-regulated expression gene, while the other five genes (ANTXR2, BAZ2B, C5AR1, PDK4, and STK3) were all up-regulated. Meanwhile, the correlation between individual hub genes and immune cells is shown in a scatter plot (Figs. S1–S8). R-value indicates positive and negative correlation. $P$-value < 0.05 is statistically significant. This evidence suggests that changes in the immune microenvironment of IS patients may be related to these six hub genes.

## Construction of ceRNA networks for hub genes

Many studies have confirmed that ceRNA regulatory networks play a role in the biology and pathophysiology of various diseases. We also constructed networks (Fig. 9, Table S7) to determine whether the six hub genes have similar regulatory relationships. The six hub genes were separately entered into the database, yielding a total of 306 miRNA-mRNA interaction pairs, of which 117 were ANTXR2-miRNA interaction pairs, 87 were BAZ2B-miRNA interaction pairs, and 285 were miRNA-lncRNA interaction pairs. We hypothesised that ANTXR2 and BAZ2B might be involved in the regulation of ceRNA. According to the value of a degree (Table S8), hsa-miR-766-3p, hsa-miR-149-3p, hsa-miR-1972, hsa-miR-186-5p, hsa-miR-1207-5p, MUC19, and LINC01002 may play a vital role in the ceRNA network.

## Establishment of drug-hub genes regulatory networks

Drugs related to the six essential genes were screened out, and three genes (BAZ2B, C5AR1, and STK3) had target drugs. The gene-drug network is shown in Fig. 10 and Table S9. We retrieved a total of 58 drugs acting on the hub gene. BAZ2B is predicted to have the most targets of drug action, and a total of 30.25 drugs targeted STK3. Only one drug targeted C5AR1.

## Validation of hub genes and diagnosis model

The RT-qPCR results showed that the expression of BAZ2B, C5AR1, PDK4, and STK3 was significantly higher in the IS group (Fig. 11 and Table S10). A diagnosis model for IS classification was established using a logistic regression algorithm based on the four key genes. The results show that the diagnostic model could distinguish patients from normal samples. The ROC curves for the hub genes and models are shown in Fig. 12A and Fig. 12B. The AUC values are as follows: BAZ2B (AUC = 0.892), C5AR1 (AUC =

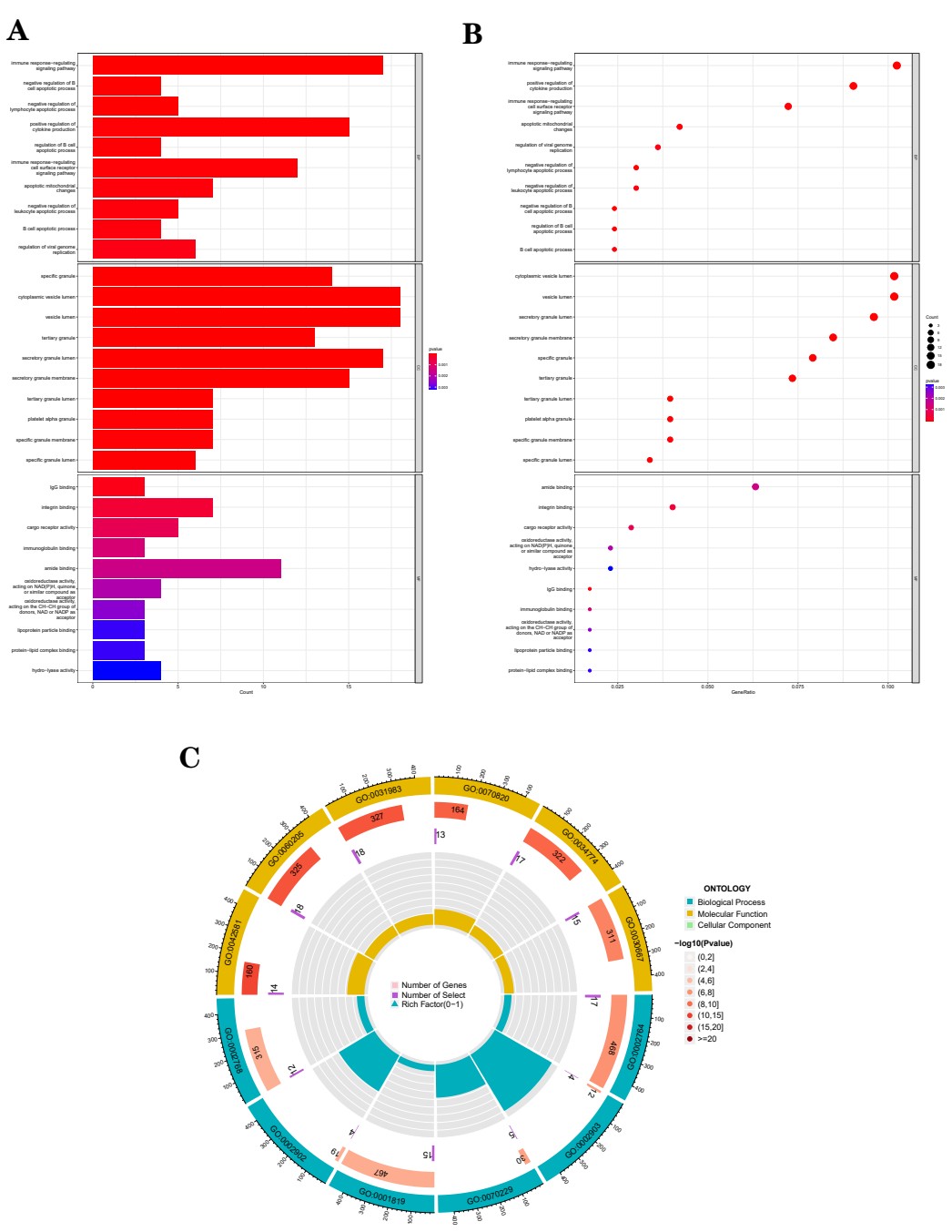

**Figure 3** **The top 10 significantly enriched biological process (BP), cellular component (CC), and molecular function (MF).** (A) In the barplot, the bar size represents the number of genes, and the colour shades represent the *p*-value. (B) In the bubble plot, the *x*-axis represents the gene ratio, the *y*-axis represents the -log10 (FDR) value, the bubble size represents the number of genes, and the colour shades represent the size of the *p*-value. (A) and (B) The *p*-value ≤ 0.001 

**Figure 3 (...continued)**
is shown in red; 0.001 < p-value ≤ 0.002 is shown in pink; 0.002 < p-value ≤ 0.003 is shown in purple; p-value > 0.003 is shown in blue. Immunoglobulin binding and amide binding are shown in pink. Oxidoreductase activity, acting on NAD(P)H, quinone or similar compound as acceptor and oxidoreductase activity, acting on the CH-CH group of donors, NAD or NADP as acceptor are shown in purple. Lipoprotein particle binding, protein-lipid complex binding and hydro-lyase activity are shown in blue. Others are shown in red. (C) The Gene Ontology (GO) enrichment analysis results of DEGs are shown as circle plots. First circle: enriched classification, outside the process, is the sitting scale for the number of genes. Different colours represent different classifications; second circle: the number of genes in the category in the background and the Q or P value. The more genes, the longer the bar and the smaller the value the redder the colour; third circle: the total number of foreground genes; fourth circle: RichFactor value for each classification (the number of foreground genes divided by the number of background genes in that classification), with each cell of the auxiliary background line indicating 0.1.

0.966), PDK4 (AUC = 0.891), STK3 (AUC = 0.966), and Model (AUC = 0.999). Finally, we verified the accuracy of hub genes (Fig. 12C) and the model (Fig. 12D) using another dataset, GSE58294. AUC values for genes and models were as follows: BAZ2B (AUC = 0.936), C5AR1 (AUC = 0.734), PDK4 (AUC = 0.697), STK3 (AUC = 0.751), and Model (AUC = 0.940). The boxplot (Fig. 13) shows the differential expression of hub genes in the CTL and IS groups in the validation dataset. Our established model performed well in distinguishing between IS and normal samples.

## DISCUSSION

IS is a disease that threatens people's health worldwide, with high incidence, disability rate, and mortality. It is urgent to develop more convenient diagnosis and treatment programs in addition to the current methods in order to help solve this problem. Primary ischemic brain injury is combined with multiple mechanisms, including excitotoxicity, oxidative stress, apoptosis, and inflammation (*Moskowitz, Lo & Iadecola, 2010*). After that, a series of immune cascade reactions can further aggravate the damage to the brain tissue, such as the production of proinflammatory cytokines and activation of destructive serine proteases (*Anrather & Iadecola, 2016*; *Xu et al., 2011*). In our research, we explored possible biomarkers of IS, mechanisms of action, and potential drug targets in terms of inflammation and immunity.

In this study, 188 DEGs were observed through R package limma. By applying the joint LASSO and SVM algorithm, we finally obtained six genes (ANTXR2, BAZ2B, C5AR1, PDK4, PPIH, and STK3) most associated with IS. However, only BAZ2B, C5AR1, PDK4, and STK3 were upregulated expression in IS patients as verified by RT-qRCR. C5AR1 has been proven to play a crucial role in regulating inflammatory and neurocognitive functions in IS, Alzheimer's disease, malaria, and neuropathic pain (*Brandolini et al., 2019*; *McDonald et al., 2015*; *Moriconi et al., 2014*). A marked increase in C5AR1 expression was observed in the MCAO- and OGD-induced models. Meanwhile, C5AR1 inhibitors have significant neuroprotective effects and notably inhibit neuro-inflammation and apoptosis in primary cortical neurons and MCAO-induced stroke models (*Brandolini et al., 2019*; *Shi et al., 2017*). TLR4 and C5AR1 promote apoptosis and inflammation by activating the cAMP/PKA/I- κB/NF- κB signaling pathway during brain ischemia-reperfusion (*Kim &*

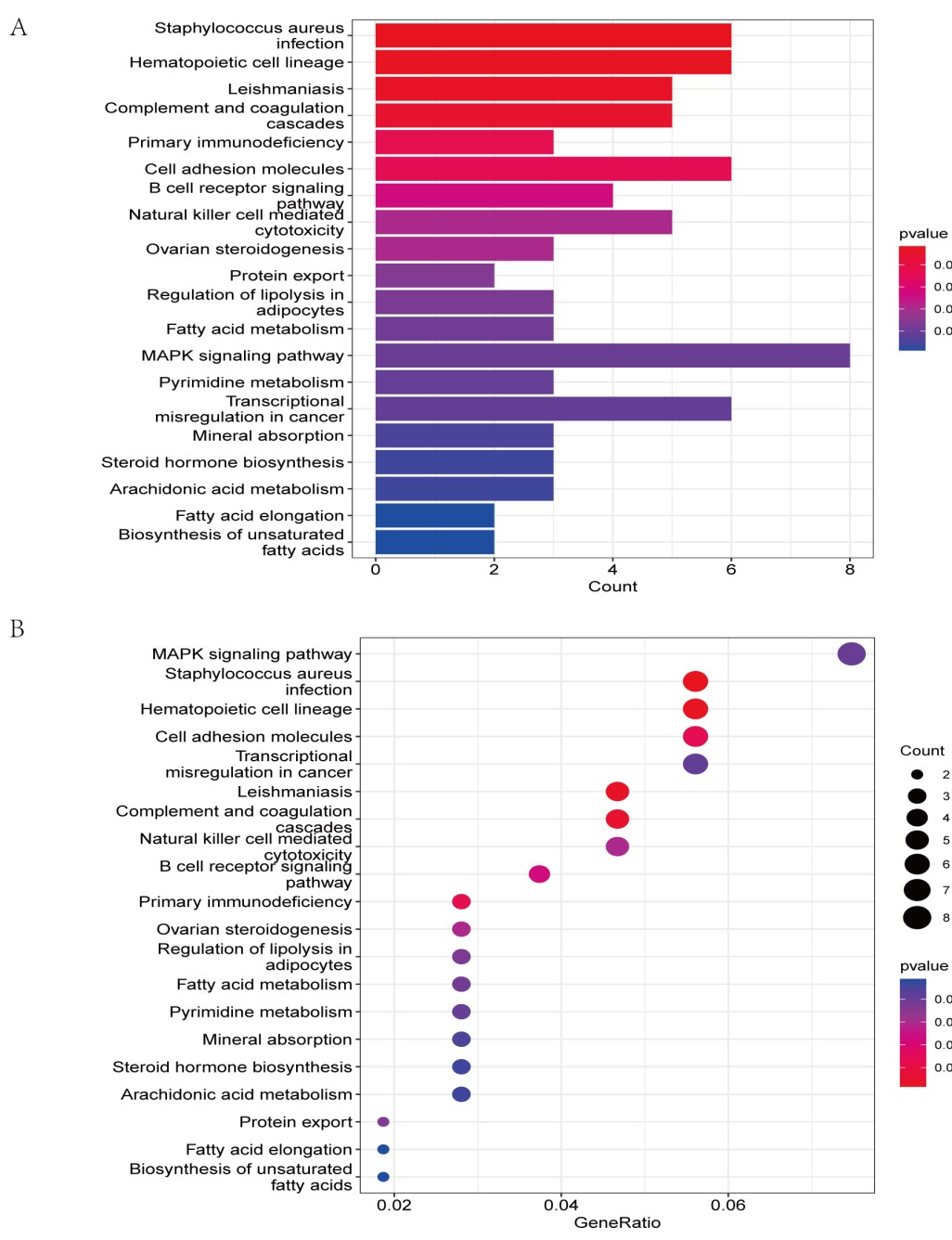

**Figure 4  The KEGG pathway analysis results of DEGs.** (A) In the barplot, the bar size represents the number of genes, and the colour shades represent the *p*-value. (B) In the bubble plot, the *x*-axis represents the gene ratio, the *y*-axis represents the -log10 (FDR) value, the bubble size represents the number of genes, and the colour shades represent the size of the *p*-value. The *p*-value ≤ 0.01 is shown in red; 0.01 < *p*-value ≤ 0.02 is shown in light red; 0.02 < *p*-value ≤ 0.03 is shown in light purple; 0.03 < *p*-value ≤ 0.04 is shown in purple; *p*-value > 0.04 is shown in blue. Staphylococcus aureus infection, hematopoietic cell lineage, leishmaniasis, and complement and coagulation cascades are shown in red. Primary immunodeficiency and cell adhesion molecules are shown in light red.

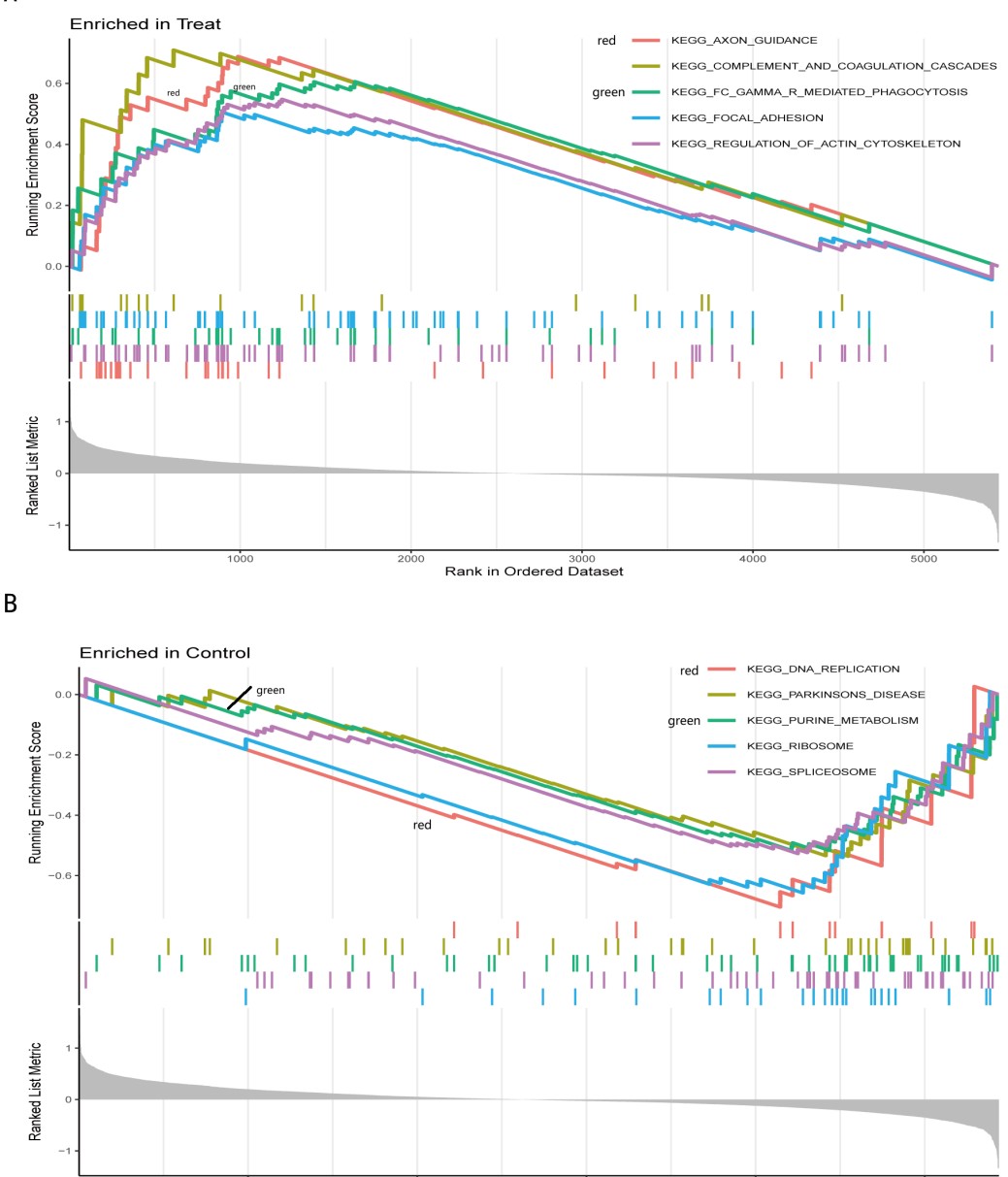

**Figure 5 Gene set enrichment analysis of IS and CTL groups in the GSE dataset.** The different colours represent the different pathways obtained by enrichment. The figure has three parts. Part I: A line graph of the gene enrichment score (ES). The vertical axis is the corresponding running ES. There is a peak at the top or bottom of the line. The core genes are those before the peak for the treatment group with a positive ES, while the core genes are those after the peak for the CTL group with a negative ES. The horizontal axis represents each gene under this gene set and corresponds to the barcode-like vertical line in the second part. Part II: Hits, each vertical line corresponds to a gene under this gene set. Part III: The distribution of rank values for all genes, and the vertical coordinate is the ranked list metric, which is the value of the rated amount of that gene. (A) The GSEA results of the IS group. (B) The GSEA results of the CTL group.

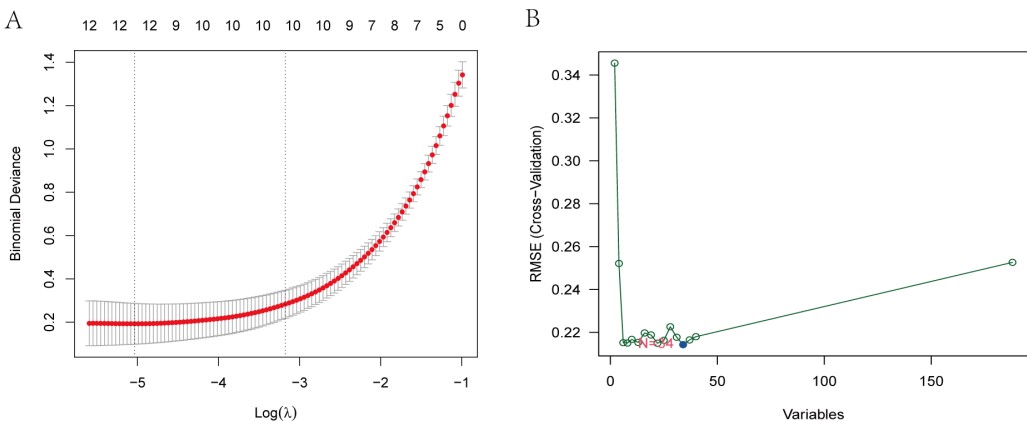

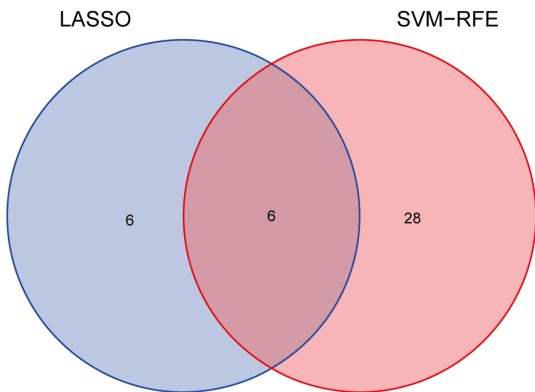

**Figure 6 The identification of six hub genes.** (A) The LASSO logistic regression algorithm identified 12 IS-related features with a 10-fold cross-validation set for selecting the penalty parameter to determine the optimal lambda value. (B) A total of 34 feature genes were filtered out using the SVM-RFE algorithm. (C) Venn diagram of genes extracted from LASSO and SVM-RFE methods. Abbreviations: least absolute shrinkage and selection operator (LASSO), support vector machine (SVM), recursive feature elimination (RFE).

*Jang, 2017*; *Shi et al., 2017*; *Zaal et al., 2017b*). Previous studies showed that C5AR1 can inhibit human monocyte-derived dendritic cells and potentially exacerbate inflammatory responses (*Zaal et al., 2017a*; *Zaal et al., 2017b*).

There have been no studies on the correlation between IS and BAZ2B, PDK4, and STK3 genes. The biological function of BAZ2B remains unclear, besides its involvement in nucleosome remodelling (*Oppikofer et al., 2017*). However, it includes at least four functional domains that could encode or bind multiple proteins or DNA to perform various roles. Due to this characteristic, we speculate that BAZ2B may be associated with many diseases, including IS. However, previous studies were limited to suggesting that it is associated with neurodevelopment, its functional loss and haploinsufficiency are one of the

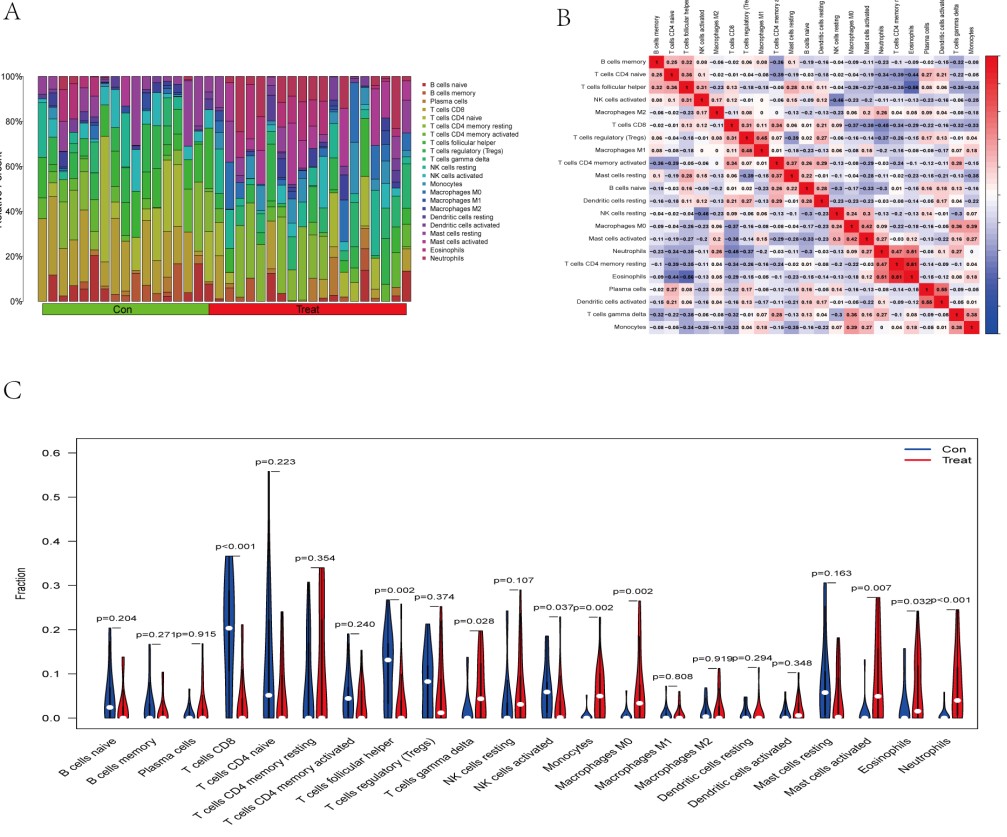

**Figure 7** **Immune infiltration landscape between the IS and CTL groups obtained by CIBERSORT analysis.** (A) The bar-plot diagram indicates the relative percentage of different types of immune cells between IS and CTL. (B) The heatmap shows the correlation in the infiltration of innate immune cells. (C) Violin plot showing the difference in immune cell infiltration between IS (red) and Control (purple), $P < 0.05$ was considered statistically significant.

causes of autism, and it may act through transcriptional regulation (*Krupp et al., 2017*; *Scott et al., 2020*). Another study found that BAZ2B activates M2 macrophages to participate in the inflammatory response (*Xia et al., 2021*), indicating its research value in immunity. PDK4 is a mitochondrial matrix enzyme essential in cellular energy regulation, which regulates the pyruvate dehydrogenase complex in the CNS and has important implications for neuron-glia metabolic interactions (*Jha, Jeon & Suk, 2012*). Atherosclerosis is the result of cholesterol and lipid deposition in the arterial wall, usually associated with calcification, and is the most common cause of IS. *Ma et al. (2020)* found that PDK4 could promote vascular calcification by interfering with autophagic activity and metabolic reprogramming, contributing to the development of atherosclerosis. STK3 (MST2) is a component of the MAPK module and Hippo signaling pathway. In cardiovascular disease studies, STK3 was found to mediate mir155 to initiate inflammation and redox stress, leading to vascular smooth muscle cell proliferation and remodeling. It regulates the ERK1/2 signaling pathway by competing with MAP2K of the MAPK pathway, which in turn triggers inflammation

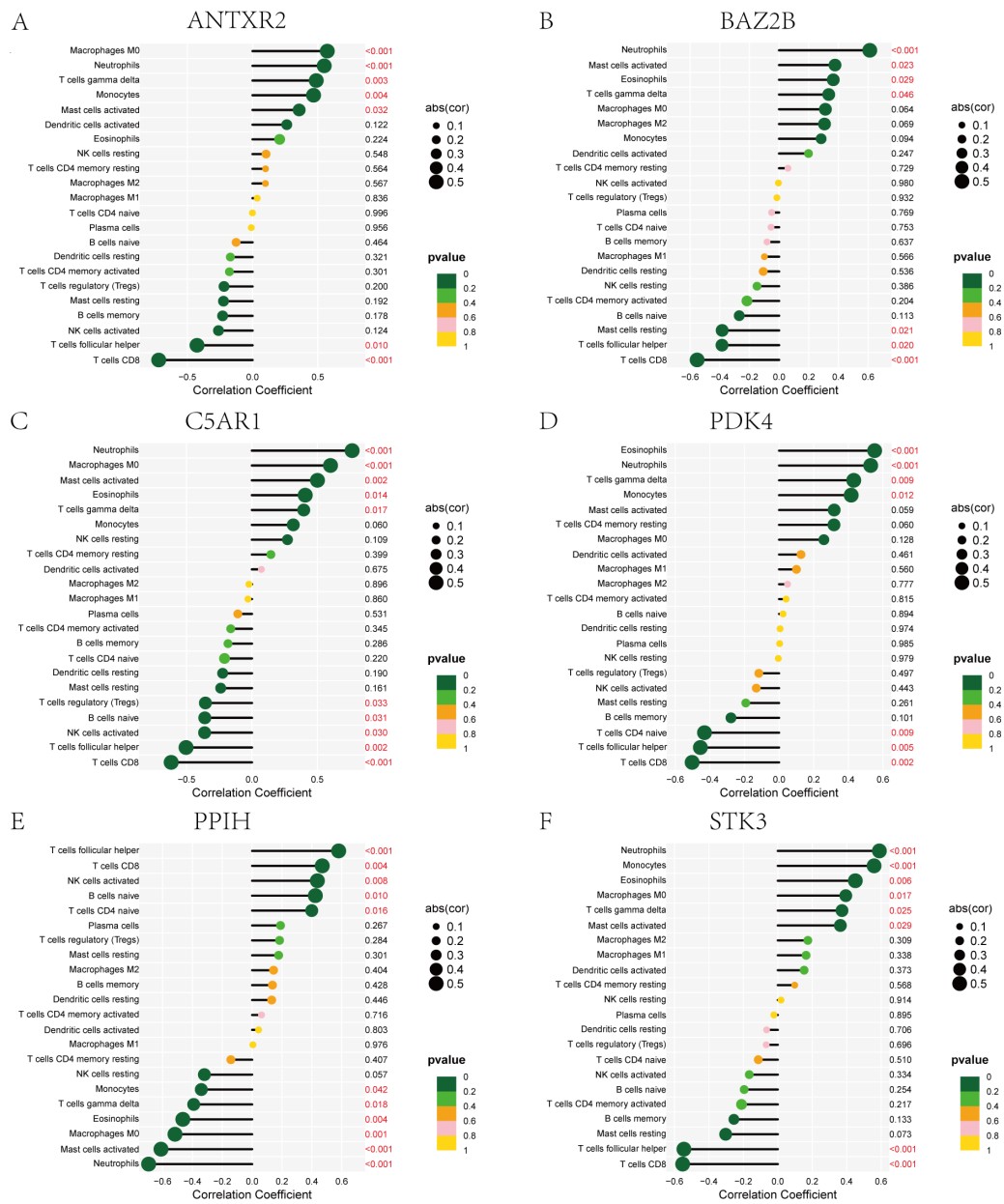

**Figure 8    Lollipop charts show the correlations between hub genes and infiltration level.** The *p*-value ≤ 0.2 is shown in green; 0.2 < *p*-value ≤ 0.4 is shown in lime green; 0.4 < *p*-value ≤ 0.6 is shown in orange; 0.6 < *p*-value ≤ 0.8 is shown in pink; 0.8 < *p*-value ≤ 1 is shown in yellow. (A) ANTXR2, (B) BAZ2B, (C) C5AR1, (D) PDK4, (E) PPIH, (F) STK3.

and oxidative stress after vascular injury (*Thiriet, 2018*). In IS, the MAPK pathway also acts by regulating cytokines, inflammation, apoptosis, and death (*Qin et al., 2022*; *Sun & Nan, 2016*). STK3 is highly expressed in most cell types in the brain and may play a role in stroke by regulating the MAPK pathway. It has also been described as a substrate for CASP6 to intervene in neurodegeneration and apoptosis (*Riechers et al., 2016*; *van Raam et*

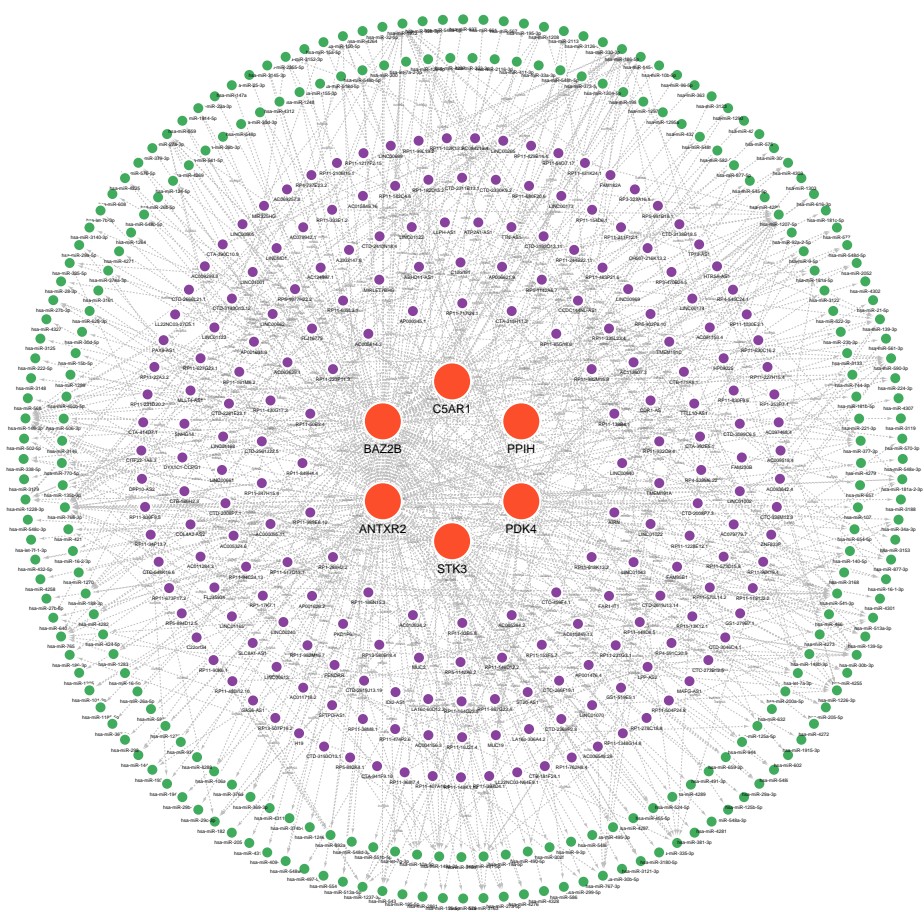

**Figure 9** **A ceRNA network based on hub genes.** The network includes nodes (six mRNAs, 249 miRNAs, 218 lncRNAs) with 591 edges. The most central part uses red orbs to represent the hub genes. The outer-most two laps are shown with green orbs representing the miRNAs, and purple orbs represent lncRNAs.

*al., 2013*). *Cho et al. (2021)* found that STK3 increases phagocytosis of adipocytes, leading to obesity due to reduced catabolic function of adipocytes. In obese human patients, STK3 expression levels were elevated. STK3 inhibitors improved metabolic patterns in obese mouse models, suggesting that there may be a viable pathway to investigate and develop drugs targeting STK3 to treat obesity-related diseases including stroke. All of the above suggest that STK3 may be relevant to the onset of IS, but the exact mechanism needs to be further investigated.

The critical pathways enriched by GO and KEGG analysis were related to immunity, including the immune response-regulating, immune response-regulating cell surface receptor, and MAPK signaling pathways. Notably, C5AR1 and STK3 were enriched in the positive regulatory pathway of the MAPK cascade. Previous studies have found that the MAPK signaling pathway in inflammation and BBB dysfunction MAPK is comprised of three main effectors: ERK1/2, JNK, and p38 (*Qin et al., 2022*; *Sun & Nan, 2016*). C5AR1 affects inflammation in bone by activating the MAPK pathway and regulating

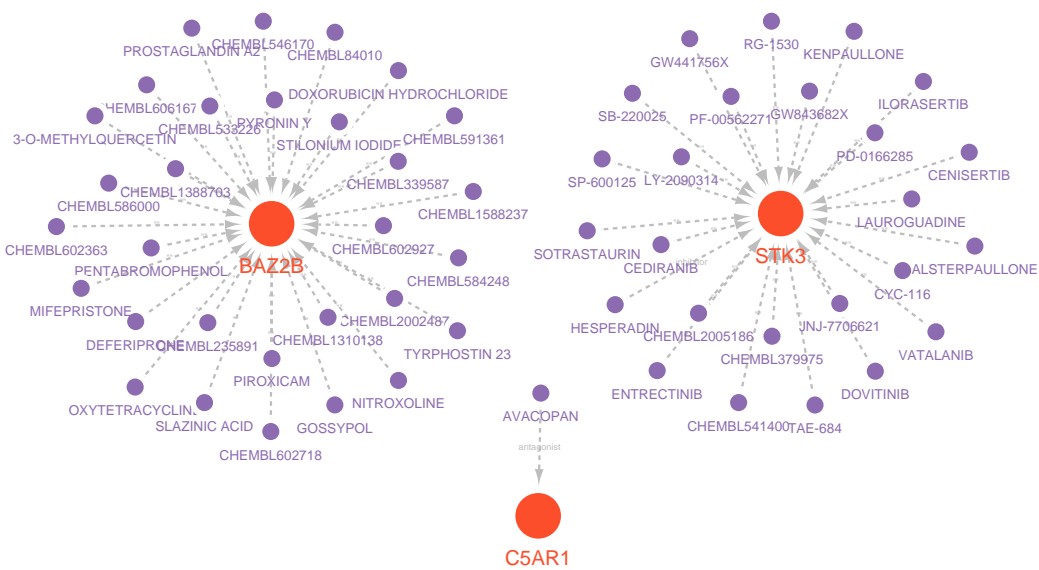

**Figure 10** **Prediction of marker gene-targeted drugs.** Red orbs represent up-regulated mRNA and purple orbs represent drugs.

gene expression in pathways associated with insulin and transforming growth factor-$\beta$ (*Modinger et al., 2018*). STK3, as described above, mediates cardiovascular injury through the MAPK pathway and also promotes apoptosis by inducing the activation of JNK (*Chen et al., 2018*). These results are consistent with our bioinformatics analysis and suggest that the MAPK signaling pathway plays an important role in these key gene-mediated biological processes. Additionally, the results of the GSEA enrichment analysis showed that C5AR1 was involved in the complement and coagulation cascades pathway, which has been proven to correlate with IS (*Berkowitz et al., 2021*). The functions of PDK4 are all focused on the involvement of lipid metabolism in the process of atherosclerosis. These results are also in accordance with the role of C5AR1 and PDK4 found in some studies (*Berkowitz et al., 2021*; *Ma et al., 2020*).

Many stroke models have confirmed that neutrophils in the acute phase of IS are among the earliest immune cells recruited into the brain tissue (*Gokhan et al., 2013*; *Jickling et al., 2015*; *Kaito et al., 2013*). Activated neutrophils may contribute to the development of IS by stimulating the systemic inflammatory response and destroying the BBB. *van Duijn, Kuiper & Slutter (2018)* found that CD8 T cells can exacerbate the inflammatory response by secreting various inflammatory cytokines, resulting in increased instability of atherosclerotic plaques. But CD8 T cells also have an anti-atherosclerotic effect, which is mediated by stimulation of inhibitory receptor production and cytolytic killing of antigen-presenting cells. The results of immunological infiltration analysis by CIBERSORT revealed that monocyte, macrophage M0, neutrophil, and mast cell activation increased, while T follicular helper and CD8 cells decreased infiltration in the IS group, which was consistent with previous studies (*Zheng et al., 2022*). Our research also showed that neutrophils were significantly and positively correlated with genes ANTXR2, BAZ2B, C5AR1, PDK4, and

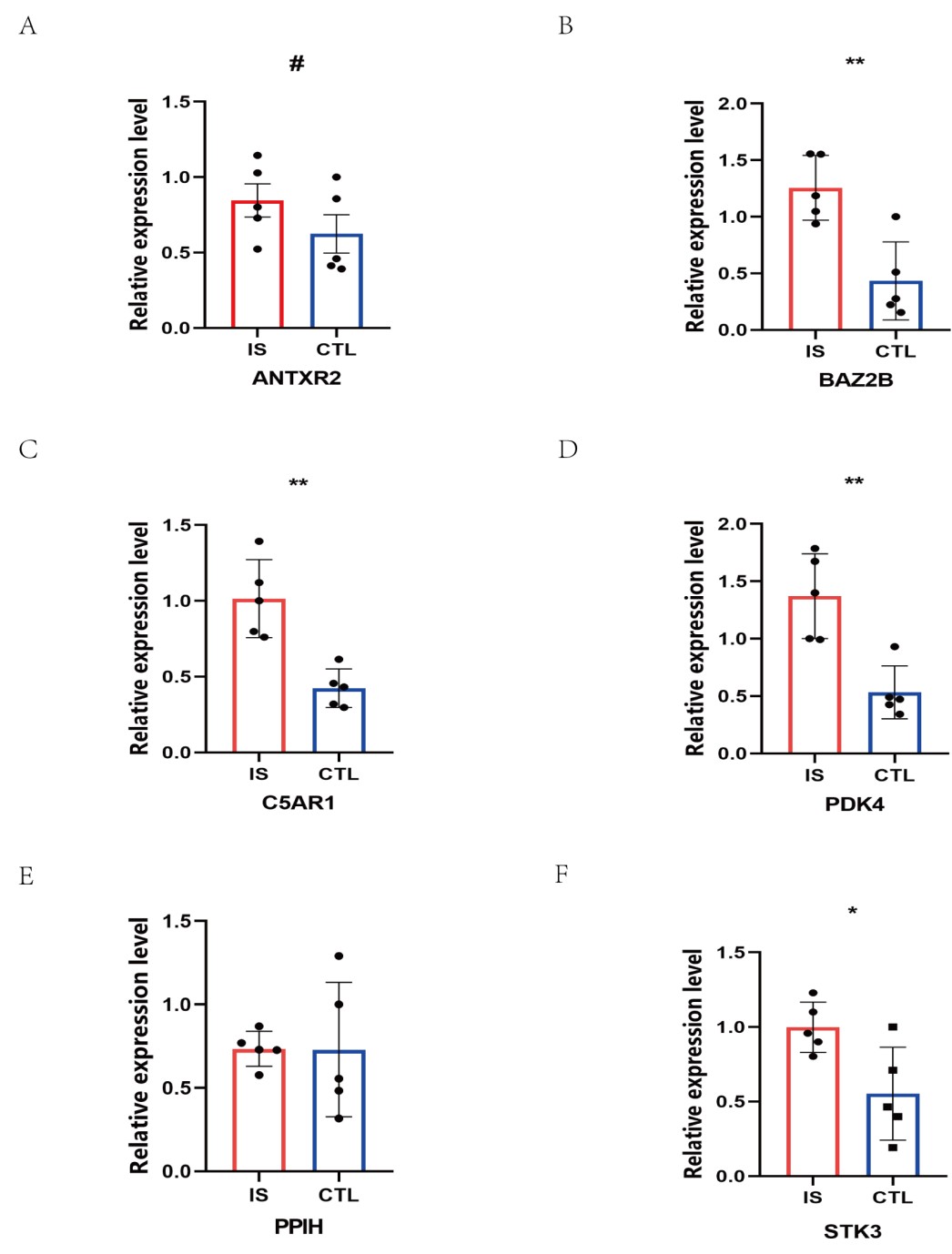

**Figure 11 Prediction of marker gene-targeted drugs.** The relative expression levels of (A) ANTXR2, $P = 0.2233$; (B) BAZ2B, $P = 0.0034$; (C) C5AR1, $P = 0.0018$; (D) PDK4, $P = 0.0026$; (E) PPIH, $P = 0.9792$; (F) STK3, $P = 0.0226$ in IS and CTL. #$p > 0.05$, *$0.01 \leq p \leq 0.05$, **$p < 0.01$. Results for the IS group are shown in red. Results for the CTL group are shown in blue.

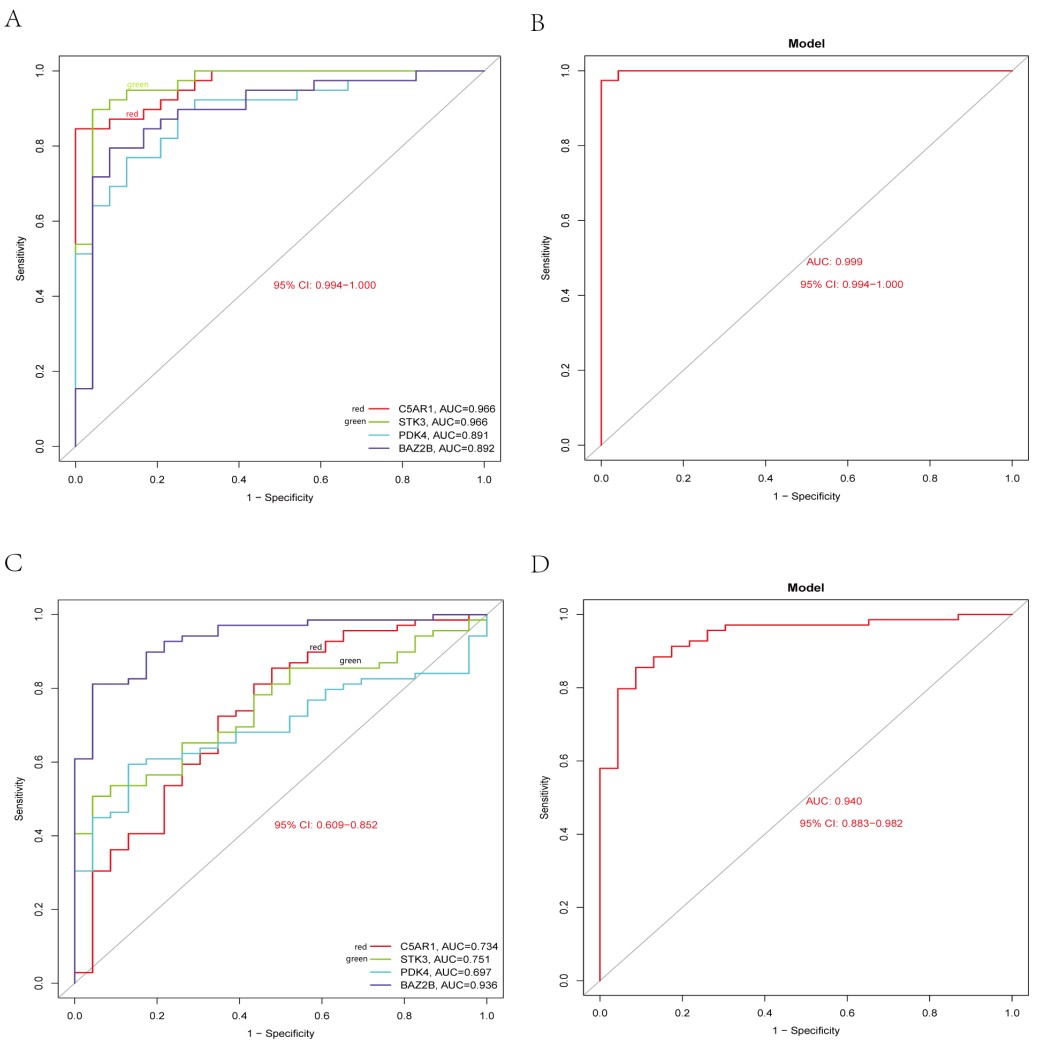

**Figure 12  Diagnostic model building and ROC curve validation.** (A) ROC curves for evaluating the accuracy of hub genes of the training set. (B) ROC curves for evaluating the accuracy of the model of the training set. (C) ROC curves for assessing the accuracy of hub genes using the validation set. (D) ROC curves for assessing the accuracy of the model using the validation set.

STK3, negatively correlated with PPIH. CD8 was significantly and negatively correlated with genes ANTXR2, BAZ2B, C5AR1, PDK4, and STK3, and positively correlated with PPIH. This evidence further suggests that neutrophils play a significant role, and hub genes are involved in regulating the immune microenvironment in IS patients.

The ceRNA network we conducted included nodes (six mRNAs, 249 miRNAs, and 218 lncRNAs) with 591 edges. In these nodes, the miRNAs with high degrees, hsa-miR-766-3p, hsa-miR-149-3p, and hsa-miR-186-5p, have been reported as essential molecules in cerebral ischemic/reperfusion injury (*Cai et al., 2019*; *Hu et al., 2020*; *You et al., 2022*). Of these miRNAs, hsa-miR-766-3p is known to be involved in immune responses or inflammation in various diseases. It contributes to anti-inflammatory responses through the indirect

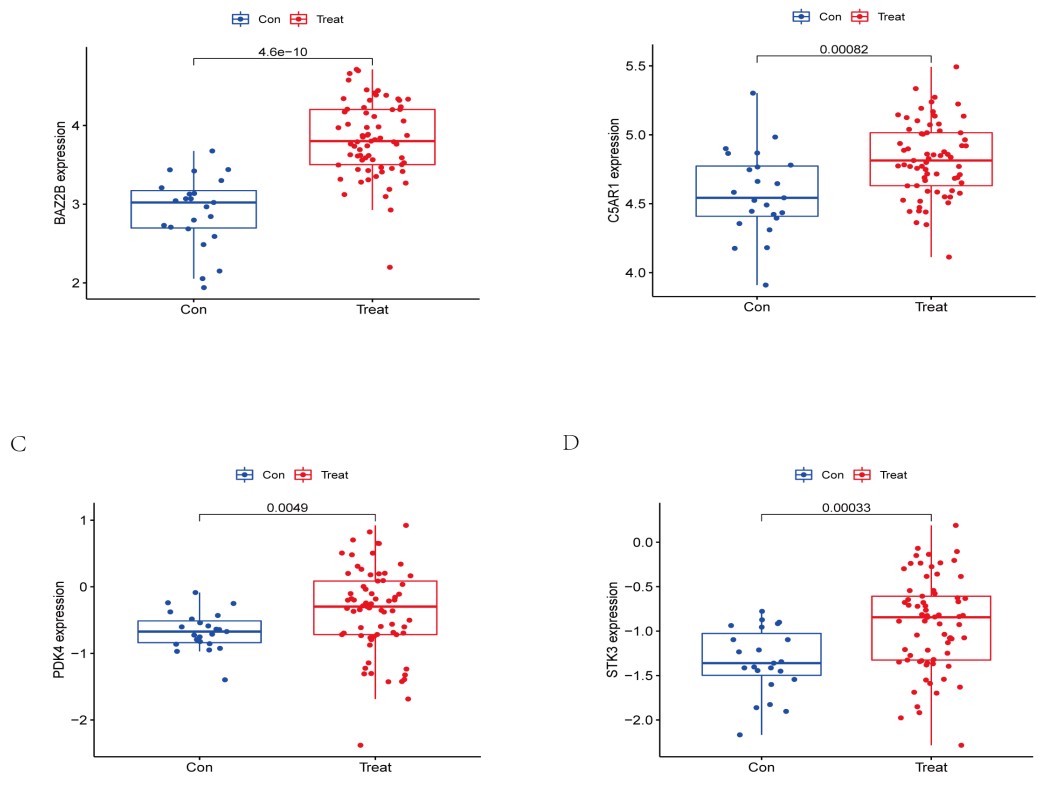

**Figure 13  External validation of the key genes.** The expression levels of hub genes (A) BAZ2B, (B) C5AR1, (C) PDK4, (D) STK3 in the GSE58294 testing set. The results for the treatment group are shown in red. The results for the CTL group are shown in blue.

inhibition of NF-$\kappa$B signaling in rheumatoid arthritis (RA) (*Hayakawa et al., 2019*). The miR-766-3p/NR3C2 axis is participating in the protection against cerebral ischemia and reperfusion (*Cai et al., 2019*). LINC00689, one of the highest degree lncRNAs in the ceRNA network, is a key regulator of the toll-like receptor (TLR) signaling pathway controlling innate immunity (*Liu et al., 2020*). However, the mode of regulation between them and hub genes still needs further validation. Meanwhile, we constructed a hub gene-based gene-drug regulatory network and predicted possible targeted drugs. The result reflected that BAZ2B had the most targets of drug action, likely due to its multiple functional areas. The network provides a theoretical basis for developing targeted immunotherapies for IS.

Finally, based on the four key genes mentioned above, we constructed an IS diagnostic model using logistic regression methods. We also verified the validity of the model in a publicly available dataset. Bioinformatics analysis combined with machine learning methods achieved good results.

However, there are some limitations to our study. First, although we analyzed clinical data and performed RT-qPCR validation, the sample size was relatively limited. Second, more *in vivo* and *in vitro* studies are needed to identify and validate the underlying mechanisms between these genes and stroke. These will be the most critical aspects of our

future research to gain insight into the immunological pathogenesis of IS and to provide additional options for clinical treatment.

## CONCLUSION

We explored six hub genes (ANTXR2, BAZ2B, C5AR1, PDK4, PPIH, and STK3) for IS and revealed the immune cell infiltration pattern *via* bioinformatics analysis and machine learning algorithm. We successfully constructed the ceRNA and drug-gene networks, which provide new ideas to explore the possible molecular mechanisms and develop targeted drugs of IS. Finally, BAZ2B, C5AR1, PDK4, and STK3 were verified to be the biological markers of IS. Based on the four genes, we developed a diagnostic model and validated its effectiveness, providing additional tools for screening and diagnosing IS.

## ACKNOWLEDGEMENTS

We want to thank the GEO database for the data support.

### Funding

This study was supported by the Research Project of Heilongjiang Provincial Health and Family Planning Commission (NO. 2018126). The funders had no role in study design, data collection and analysis, decision to publish, or preparation of the manuscript.

### Grant Disclosures

The following grant information was disclosed by the authors:
Research Project of Heilongjiang Provincial Health and Family Planning Commission: 2018126.

### Competing Interests

The authors declare there are no competing interests.

### Author Contributions

- Lin Cong conceived and designed the experiments, prepared figures and/or tables, authored or reviewed drafts of the article, and approved the final draft.
- Yijie He conceived and designed the experiments, prepared figures and/or tables, and approved the final draft.
- Yun Wu analyzed the data, prepared figures and/or tables, and approved the final draft.
- Ze Li performed the experiments, prepared figures and/or tables, and approved the final draft.
- Siwen Ding performed the experiments, authored or reviewed drafts of the article, and approved the final draft.
- Weiwei Liang analyzed the data, prepared figures and/or tables, and approved the final draft.

- Xingjun Xiao performed the experiments, authored or reviewed drafts of the article, and approved the final draft.
- Huixue Zhang analyzed the data, authored or reviewed drafts of the article, and approved the final draft.
- Lihua Wang conceived and designed the experiments, authored or reviewed drafts of the article, and approved the final draft.

## Human Ethics

The following information was supplied relating to ethical approvals (i.e., approving body and any reference numbers):

The Medical Ethics Committee of the Second Affiliated Hospital of Harbin Medical University approval to carry out the study within its facilities (NO. KY2022-285).

## Data Availability

The data is available at NCBI GEO: GSE16561, GSE58294.

The raw data is available in the Supplementary Files, Github, and Zenodo:

– https://github.com/donkeycong/Ella.git.

– Cong, L. (2024). Discovery and validation of molecular patterns and immune characteristics in the peripheral blood of ischemic stroke patients. In Discovery and validation of molecular patterns and immune characteristics in the peripheral blood of ischemic stroke patients. Zenodo. https://doi.org/10.5281/zenodo.10869350.

## Supplemental Information

Supplemental information for this article can be found online at http://dx.doi.org/10.7717/peerj.17208#supplemental-information.

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
