# Peer review of "Discovery and validation of molecular patterns and immune characteristics in the peripheral blood of ischemic stroke patients"

_PeerJ, doi:10.7717/peerj.17208_

## Round 0.1 · original submission · Major Revisions

Our reviewers have a variety of suggestions for you, ranging from checking details to overall writing quality and to the need for a more thorough explanation and discussion of your methodology, conclusions, and train of thought. Please address them all thoroughly

Reviewer 1 ·

Basic reporting

- Many references were missing. E.g., Cassella and Jagoda, Malik, Torres-Aguila, Wu, etc.

- Labels in Fig3AB are blurry.

Experimental design

The aim described in the Abstract is unclear. The authors pinpointed that “the exact mechanism [of stroke] is not clear”. Then they aimed to “identify immune-related biomarkers and evaluate the infiltration pattern of immune cells”. It seems the logical connection was missing between the disease mechanism and biomarkers.

Regarding the study’s goal - “the immune factor plays a crucial role in the progression of ischemic stroke” #17-18. It is unclear what they meant by “progression of ischemic stroke”. Does it mean the imminent risk of recurrent stroke? Does it mean the deterioration of overall health or just the immune system?

In #126-136, could you describe how long after IS did these two studies collect blood samples from the patients? Were the patients suffering from the 1st stroke or recurrent stroke? These clarifications are important in understanding if the datasets can answer the research question about “the progression of ischemic stroke”.

The authors did not mention if they had mitigated or tackled the multicollinearity issue before performing LASSO. If genes that share highly correlated expression patterns are not handled, it will skew the LASSO model. Then the so-called hub genes may be due to multicollinearity.

Validity of the findings

I am confused about how Fig 6A in #257 can be used to identify the 12 hub genes.

I am curious about why the highly differentially expressed gene ARG1 in the volcano was not selected as a hub gene. Are there any biological reasons?

Diagnosis model. I am not convinced that the approach of using hub genes can be turned into a diagnosis model. Probably, there are many ailments that can perturb the hub genes. Therefore, the model, at best, can only tell people are not healthy due to various reasons in the event that the hub genes show an elevated level of expression.

Based on the volcano plot, PPIH was downregulated vs. control. I am wondering why PPIH depicted a similar correlation pattern as the other 5 hub genes in Fig8.

Reviewer 2 ·

Basic reporting

In this study, six genes associated with ischemic stroke (ANTXR2, BAZ2B, C5AR1, PDK4, PPIH and STK3) were screened out by mining GEO database. Bioinformatics analysis and machine learning were used to reveal the immune regulatory mechanisms related to the target genes. The gene-drug target network and the mRNA-miRNA-lncRNA regulatory network were constructed. RT-qPCR showed that the expression levels of four target genes (BAZ2B, C5AR1, PDK4, and STK3) in peripheral blood of patients with ischemic stroke were all higher than those of healthy people, thus establishing the corresponding diagnostic model. It provides a new idea for the screening and diagnosis of ischemic stroke and has a good innovation.

Experimental design

In this study, except for the research of C5AR1 gene in ischemic stroke, the biological function of other target genes in ischemic stroke is still unknown. Whether these genes can be used as biomarkers of ischemic stroke still needs to be supplemented in vivo and in vitro experiments.

Validity of the findings

The title of this paper is "Discovery and verification of peripheral blood molecular patterns and immune characteristics in ischemic stroke". In this study, the authors failed to explain that the constructed gene-drug target network and the mRNA-miRNA-lncRNA regulatory network are related to the immune mechanism after ischemic stroke, so it is suggested to add relevant content.

·

Basic reporting

[1] I recommend that the author of this manuscript seeks assistance from a fluent English speaker to address any English-related issues present in the text.

L144 +
We not only annotated the function of the differential expression genes themselves, Gene Ontology (GO) enrichment, but we also studied the pathway by Kyoto Encyclopedia of Genes and Genomes (KEGG) analysis.
==>
We not only annotate the function of the differentially expressed genes themselves using Gene Ontology (GO) enrichment but also studied the pathways through Kyoto Encyclopedia of Genes and Genomes (KEGG) analysis.

L150+
The R package clusterProfiler also conducted the GSEA analysis of the IS and CTL group, adjusted P-value<0.05.
==>
"The R package clusterProfiler also conducted the GSEA analysis of the IS and CTL groups, with an adjusted P-value<0.05."

[2] A lot of references were found to be missing in the manuscript, including L109–112, L131, L141, L142, L159, L180, and others.

[3] I am confused about which GSE datasets were used, is it GSE58264 or GSE58294? GSE 58264 is the Regulation of gene expression by light in the fungus Mucor circinelloides

[4] Figure 6 and Figure 7 have been misplaced and should be referenced to pages 34 to 36 of the PDF file.

[5] The descriptions of the x-axis and y-axis in Figure 11 appear to be inconsistent.

The presence of these issues suggests that there may have been some oversight or lack of attention to detail during the preparation of this manuscript by the authors.

[5] To enhance reader comprehension, it is better to divide the introductory paragraph into multiple smaller paragraphs. In the first paragraph, introduce the topic of ischemic stroke (IS), and in the subsequent paragraphs, delve into its molecular mechanism, etc. This restructuring would facilitate a clearer understanding for the readers.

Experimental design

I have no objections to the utilization of standard bioinformatics analysis methods presented in this paper.

The authors applied two machine learning algorithms (LASSO and SVM-REF) to identify the hub genes and used CIBERSORT to analyze the immune cell infiltration pattern. Then, they built Gene-Drug target networks and mRNA-miRNA-lncRNA regulatory networks with Cytoscape. To confirm the hub genes, they performed RT-qPCR and developed diagnostic models using logistic regression methods.

Validity of the findings

The study explores the immune-related biomarkers and immune cell infiltration patterns in ischemic stroke patients. It reveals the specific molecular patterns and immune characteristics and confirms that four of the six DEGs were significantly upregulated in ischemic stroke patients by RT-qPCR.
In conclusion, the authors suggest that more in vivo and in vitro studies are needed to identify and validate the underlying mechanisms between these genes and stroke, which will be critical aspects of future research. The authors anticipate that this study will provide valuable insights into the immunological pathogenesis of ischemic stroke and offer potential avenues for exploring immunomodulatory therapy options in the future.

---

## Round 0.2 · Minor Revisions

The original manuscript was reviewed by three referees, all of whom recommended Major Revision. The revised manuscript was examined by only one of the three referees. Based on their comments and my perusal of the manuscript, your manuscript is acceptable for publication provided that it is edited to correct grammar and other language issues. In its current state, the manuscript is difficult to read with clarity. I hope you can avail of a professional language editing service for this purpose.

Reviewer 1 ·

Basic reporting

Much improved.

Experimental design

Significantly improved.

Validity of the findings

No issues.

Additional comments

The authors had answered all my questions.

---

## Round 0.3 · accepted · Accept

Language & grammar have been extensively revised, making it easier to understand the manuscript.